# Perspectives on the Management of Oligometastatic Disease in Esophago-Gastric Cancer

**DOI:** 10.3390/cancers14215200

**Published:** 2022-10-23

**Authors:** Thorsten Oliver Goetze, Salah-Eddin Al-Batran

**Affiliations:** Krankenhaus Nordwest gGmbH, Institut of Clinical Cancer Research, UCT—University Cancer Center Frankfurt-Marburg, Steinbacher Hohl 2-26, 60488 Frankfurt, Germany

**Keywords:** gastric cancer, gastroesophageal cancer, limited metastatic disease, oligometastatic disease, chemotherapy, immunotherapy, peritoneal carcinomatosis, Peritoneal Cancer Index

## Abstract

**Simple Summary:**

In metastatic stages of gastric cancer, surgical intervention with curative or life-prolonging intention is still a highly individualized as well as experimental approach, which is unfortunately often performed in daily routines without the sufficient data of well-powered randomized trials. In particular, a small subgroup of metastatic patients showing a limited number of distant metastases in a restricted number of organs, termed limited or oligometastatic disease, needs further evaluation and is discussed in the current manuscript.

**Abstract:**

Gastric adenocarcinoma and esophageal cancer are the fifth and seventh most common cancer types worldwide. At the time of initial diagnosis, up to 50% of esophagogastric cancers present with distant metastatic lesions and are candidates for chemotherapy. Curative surgery in this stage is still an experimental approach. Only a small number of these metastatic patients show an oligometastatic disease with no uniform definition of what oligometastatic means in gastric cancer. Nevertheless, the question remains unanswered as to whether these patients are still candidates for curative concepts. Some studies have attempted to answer this question but have not been adequately designed to address the role of a curative-intended multimodal therapy in this setting. The current FLOT-5 is designed to potentially provide a definitive answer to the question of whether curatively intended surgery plays a role or is a disadvantage in this setting.

## 1. Introduction: General Aspects of Oligometastatic Esophagogastric Cancer

Gastric adenocarcinoma and esophageal cancer, with an estimated 1,033,701 and 572,034 new cases annually, are the fifth and seventh most common cancer types, respectively, worldwide [1]. At the time of initial diagnosis, up to 50% of esophagogastric cancers present with distant metastatic lesions and are candidates for chemotherapy [2,3]. Also, up to 50% of patients with local or local advanced disease develop metachronous metastatic disease in the further course of their cancer disease [4,5,6].

In metastatic stages of gastric cancer, surgical intervention with curative or life-prolonging intention is still a highly individualized as well as experimental approach. It has been evaluated in several trials with a limited number of patients only and subgroup analyses of studies as well as retrospective patient cohorts. Only a small number of these metastatic patients show distant metastases with a limited number of lesions in a limited count of organs: so-called limited or oligometastatic disease [7].

Obtained data indicate that surgery could possibly provide a benefit for a highly selected subgroup such as, e.g., patients aged 70 years or less with one metastatic site only (lymph nodes or liver) [8] and an excellent response to induction preoperative chemotherapy [9,10], or patients with metastases limited to the liver, in whom a complete liver resection of all metastases is possible [11]. Nevertheless, there are no hard and fast criteria or sufficient data for these often individual decisions.

A research group from Germany identified D3 lymph nodes as an independent predictor for overall survival (OS) based on the data of 48 gastric cancers with gastric primary resection [12]. In addition, colleagues from Japan investigated a cohort of 16 gastric cancer patients with positive para-aortal lymphatic node disease who received surgery with curative intent after induction with a taxane, platinum and fluoropyrimidine triplet systemic therapy and demonstrated a promising two year survival rate of 93.8% as well as a relapse-free survival (RFS) rate of 75.0%. [13]. These data were supported by several case reports in highly selected patients indicating a possible benefit from these multimodal concepts [14,15,16].

The concept of limited/oligometastatic disease (OMD) was introduced in 1995 and described a clinical disease state of only limited metastatic burden [17]. In 2020, a comprehensive classification system for OMD was proposed by The European Society for Radiotherapy and Oncology (ESTRO) and The European Organization for Research and Treatment of Cancer (EORTC) as a consensus recommendation [18]. The definition of the consensus recommendation characterized different scenarios. Here, a distinction is made between a previously already existing diffuse metastatic disease, in which a limited metastatic disease is induced by therapy, and a genuine so-called upfront limited or oligometastatic disease. The genuine limited metastatic disease is further subclassified into repeated OMD (already an existing history of limited metastatic disease) and de novo OMD, which are further subdivided into synchronous or metachronous OMD. Furthermore, there is a differentiation into oligorecurrence, olgioprogression, and oligopersistence, to underline if the OMD occurred during a treatment-free interval or under an ongoing course of chemotherapy and whether an oligometastatic lesion is progressing on current imaging [18]. As of now, no uniform criteria exist regarding the maximum number of lesions and organs to be considered OMD in esophagogastric cancer. Most of the trials that investigated OMD defined it as a maximum of three lesions in one organ [19,20,21]. However, as for the consensus recommendation of the OMD classification system by ESTRO and EORTC, these classifications still need to be prospectively evaluated. Therefore, surgery in metastatic gastric cancer remains highly debatable since results of sufficiently powered randomized trials are lacking.

Figure 1 shows useful questions that should be addressed to further define patients that could be oligometastatic.

## 2. Trials in Oligometastatic Esophagogastric Cancer

A pilot of the German Gastric Group (FLOT study group) [22] established a clinical model to identify and harmonize characteristics of patients that could potentially benefit from surgical intervention in their metastatic gastric cancer disease after an initial induction systemic chemotherapy. One of the best evaluations currently available for defining OMD in gastric and esophagogastric junction (GEJ) adenocarcinomas, supported by comparatively sufficient study data, is based on the results of the German prospective multicenter FLOT-3 trial [22], which is also used in the resulting German follow-up multicenter randomized FLOT-5/RENAISSANCE trial [18,23] and in the German S3 Guidelines [24].

In the FLOT-3 trial [22] of the German Gastric Group, therapy-naïve patients with gastric or GEJ adenocarcinoma were stratified into three groups (A–C): (A) clearly resectable without evidence of systemic disease, (B) limited metastatic, and (C) patients with diffuse metastatic disease, using a previously determined algorithm and using FLOT as the standard regimen. The limited metastatic group included patients with metastatic lymph node involvement at distant intra-abdominal lymph nodes only or/and a maximum of one metastatic organ system affected (with less than five liver metastases if the affected organ was the liver), and with no signs of macroscopically detectable carcinomatosis (peritoneum or pleura) and good performance status (ECOG ≤ 1). Metastatic patients not fulfilling the above-mentioned criteria were considered as diffuse metastatic patients [22]. Within FLOT-3, patients without signs of systemic disease received induction with four cycles of FLOT triplet regimen, then radical curatively intended tumor surgery followed by four cycles of adjuvant FLOT, in accordance with the FLOT-4 landmark trial that followed later based on this concept [25,26]. In a similar manner, patients classified as limited metastatic received four FLOT regimens followed by surgical resection of all tumor sides including the primary tumor as well all metastatic sides, if possible. If surgery was possible and carried out, four adjuvant cycles of FLOT therapy were administrated. Patients with diffuse metastases received eight cycles of FLOT regimen without curatively intended surgery. Overall, 25% of the patients (*n* = 60/238) showed limited metastatic disease and more than half of them (*n* = 36) underwent surgery after induction with four cycles of FLOT regimen. These patients with surgical resection showed a survival benefit with a median OS of 31 months compared with only 16 months in the same population but without curatively intended surgery [22]. These results of FLOT-3 patients with limited metastatic disease then provided the rationale for the FLOT-5/RENAISSANCE trial. In the FLOT-5 trial, the definition used for limited/oligometastatic status was based on those used in the FLOT-3 with modifications [Table 1] [22,23].

The FLOT-5 trial was a prospective, multicenter, randomized, investigator-initiated phase III study to evaluate the effect of induction FLOT chemotherapy in chemo-naïve oligometastatic gastric or GEJ adenocarcinoma patients without prior resection of the primary or of the metastases. For HER2-positive and PD-L1-positive (combined positivity score ≥ 5) trastuzumab and nivolumab, respectively, were added to the FLOT regimen. The systemic treatment was combined with curatively intended gastrectomy/esophagectomy and resection of all metastatic lesions or local ablation/radiation procedures with the goal to achieve a R0- situation at all sites. Due to the randomized nature of the study, after 4 cycles of FLOT without signs of progressive disease, patients were randomized in a 1:1 fashion into Arm A with surgery following adjuvant systemic treatment or Arm B without surgery, the standard arm with continuation of systemic therapy only [23]. All patients of the RENAISSANCE trial were and still are centrally reviewed by an expert panel (s. flow chart of FLOT 5- trial) [Figure 2].

Another highly published trial in the OMD arena was the randomized Asian phase III REGATTA trial [27] that enrolled gastric adenocarcinoma patients with a single non-curable factor (liver, peritoneum or para-aortic lymph nodes) to chemotherapy alone or gastrectomy only followed by chemotherapy [27]. In contrast to the FLOT-3 results, this trial failed to show improvements in survival for surgically treated patients. The results even showed a trend towards inferiority in the surgery group (median OS was 16.6 months in patients without versus 14.3 months in patients with gastrectomy). However, the design of the study is highly debatable, as gastrectomy was restricted to D1 lymphadenectomy without resection of any metastatic lesions, and the important claim for cure within the trial was not given within the surgical arm.

## 3. Role of Peritoneal Metastases in Esophagogastric Cancer

Peritoneal carcinomatosis in gastric cancer is a special issue and should also be addressed in OMD trials, e.g., RENAISSANCE.

In individual cases, patients could possibly benefit from an interdisciplinary, coordinated, multimodal therapy if the following conditions are present: synchronous peritoneal metastases, isolated peritoneal involvement with a Peritoneal Cancer Index (PCI) ≤ 6 at laparoscopy, induction with neoadjuvant chemotherapy and a high probability of complete macroscopic cytoreduction.

Numerous cohort studies are available on this. In a well-selected group of patients with peritoneal metastases, a French study demonstrated a long-term remission rate of 11% after resection and hyperthermic intraperitoneal chemotherapy (HIPEC). In multivariate analysis, a PCI < 7 synchronous metastasis and achievement of complete cytoreduction were shown to be independent prognostic factors [28]. Results of a retrospective French study with 159 patients indicated that a therapeutic approach combining cytoreductive surgery (CSR) with intraperitoneal chemotherapy may achieve long-term survival in a highly selective group of patients with limited and resectable peritoneal carcinomatosis. However, the high mortality rate observed in this study underscored the need for strict selection that should be reserved for experienced institutions involved in the surgical therapy of peritoneal seeding in gastric cancer [29]. To comply with this, the currently running German PREVENT (FLOT9) study with prophylactic HIPEC, for example, is only performed at selected high-volume HIPEC centers with expertise in the field [30].

A prospective phase III trial from China showed a significant increase in median survival from 6.5 months for CSR alone to 11 months for CSR plus HIPEC (*p* = 0.046). A multivariate analysis performed on the 68 patients of this study suggested that ≥6 cycles of systemic chemotherapy, synchronous metastasis, complete cytoreduction and the absence of postoperative complications were indicative of a more favorable outcome [31]. In addition, a Japanese study evaluating bidirectional, systemic plus intraperitoneal chemotherapy with docetaxel and cisplatin followed by S-1 (tegafur/gimeracil/oteracil), prior to CSR plus HIPEC, achieved a median survival of 15.8 months. Multivariate analysis revealed the following independent prognostic factors: PCI ≤ 6, good histological response and complete cytoreduction [32].

The results of a German retrospective study including 38 patients revealed an increased median survival for peritoneal metastases patients treated with multimodal therapy including HIPEC compared with systemic chemotherapy alone (17 months vs. 11 months) [33].

In addition, a meta-analysis of prospective randomized trials (albeit almost exclusively Asian trials in Asian patients, most with fewer than 80 patients) from 1985 to 2016 showed survival benefits for patients with isolated peritoneal metastases after resection and HIPEC. Patients with positive peritoneal cytology or even those with lymph node metastases appeared to benefit from therapy [34]. However, data from Asian patients cannot be easily extrapolated to Western populations. The currently recruiting Dutch phase III PERISCOPE II trial evaluating the efficacy of CSR plus HIPEC versus palliative systemic chemotherapy in patients with peritoneal dissemination will hopefully reveal more data for central European patients. PERISCOPE II is recruiting resectable, cT3-cT4 gastric cancer patients with limited peritoneal seeding (PCI < 7) and/or tumor-positive peritoneal cytology [35].

In gastric carcinoma, there has been a long debate as to whether diagnostic techniques can reliably detect a positive endoperitoneal cytologic finding, and, even more important, what this means for prognosis and how it influences the therapeutic strategy [36].

The urgent need to increase the survival of patients with an advanced disease status led to the suggestion of intraperitoneal chemotherapy even in patients with positive lavage cytology without macroscopic peritoneal metastasis (Cy+/PC−). The impact of a combination of peritoneal lavage to wash out tumor cells and intraperitoneal chemotherapy in those patients was described in a meta-analysis. The results showed that the OS of patients was increased by intraperitoneal chemotherapy and that peritoneal lavage further increased these survival rates accompanied with a decrease in the peritoneal recurrence rate [37]. Nevertheless, extensive peritoneal lavage (EIPL) after curative gastrectomy for gastric cancer, investigated within the EXPEL trial, failed to reduce the risk of peritoneal recurrence and did not improve survival after surgery [38].

As mentioned by the CYTO-CHIP investigators, positive peritoneal cytology after gastrectomy corresponds to an R1 resection, with an equivalently bleak prognosis [39].

Based on these data, the results obtained in patients with tumor-positive peritoneal cytology without macroscopic signs of peritoneal seeding within the ongoing PERISCOPE II will be of special interest.

Interestingly a retrospective single-center cohort analysis from the University Hospital of Zurich (*n* = 172) was able to show that conversion of CY+ to CY− after neoadjuvant FLOT–chemotherapy was prognostic regarding overall survival [40].

Focusing on macroscopic peritoneal seeding in limited metastatic disease, the GYMSSA trial [41] showed that selected patients with gastric carcinomatosis and limited disease burden can achieve prolonged survival with maximal cytoreductive surgery plus regional HIPEC and systemic chemotherapy. However, GYMSSA was a single-institution trial with only 16 evaluable patients [41].

A propensity score analysis within the CYTO-CHIP study [39] showed that HIPEC plus CRS improved OS and RFS in gastric cancer with limited peritoneal metastasis without an increase in postoperative mortality or morbidity compared to CSR alone. Therefore, CRS plus HIPEC may be considered as valuable therapy for strictly selected patients with limited intraabdominal spread with a median PCI of 6, which is currently defined as the upper limit for reasonable curative CRS plus HIPEC [39].

In summary, the current data provide several indications of a possible improvement in prognosis with peritonectomy and HIPEC for limited peritoneal carcinomatosis. However, the data situation does not appear sufficient to recommend peritonectomy and HIPEC outside of clinical trials at this time. The benchmark of PCI ≤ 6 for the definition of oligometastatic disease based on the shown data is already used in the ongoing FLOT-5 trial [23].

## 4. Further Factors with Potential Influence on Oligometastatic Esophagogastric Cancer

Non-curatively treated patients with incurable tumors from the Dutch Gastric Cancer Trial [8] were studied to define more accurately which patients might benefit from palliative resection. Twenty-six percent of the randomized pts were found to have incurable tumors at laparotomy. These patients had either explorative laparotomy or gastroenterostomy or underwent resectional surgery. In general, OS was greater if resection was performed (8.1 vs. 5.4 months; *p* < 0.001). In particular, the group with only one metastatic site showed a clinically significant survival improvement due to surgical resection (10.5 vs. 6.7 months; *p* = 0.034) compared with the group with two or even more metastatic sites (5.7 vs. 4.6 months; *p* = 0.084). The maximum benefit from surgical resection was detected in the subgroup of patients younger than 70 years combined with only one metastatic site, indicating that further factors besides the tumor burden will influence the outcomes of OMD gastric cancer patients [8].

A multicenter study in patients with esophagogastric cancer and oligometastatic disease, performed in Dutch as well Swiss cancer centers, was able to show that the combination of local and systemic treatments in the limited metastatic situation was associated with an improvement in OS compared with chemotherapy alone. However, there was a mix of histology and localization with 73% esophageal cancer including 79% adenocarcinomas and metachronous seeding in half of the population indicating the need of a clearly defined characterization of this special subgroup [42]. Similarly, radiotherapy with or without chemotherapy was shown as an efficacious treatment option for a selected group of esophageal squamous cell carcinomas with 3 or fewer metastases, defined as oligometastatic, and a controlled primary malignancy after radical treatment, with all metastatic lesions amenable to stereotactic body radiation therapy (SBRT) [43]. In addition, a Dutch single-center, retrospective cohort study in oligometastatic esophagogastric cancer was able to show that OS after local treatment alone was 17 months (95% CI 12–40), after systemic therapy alone 16 months (95% CI 11–NA) and for the combination of local plus systemic therapy even 35 months (95% CI 29–NA). Here, 67% of the patients in the combination group had SBRT, 17% had metastasectomy and 17% had chemoradiation as local treatment [44]. These results underline the import role and the potency of radiation as a useful therapy in OMD esophageal cancer. In particular, the combination of local plus systemic therapy seems to be a promising approach.

Across multidisciplinary tumor boards, oligometastatic esophagogastric cancer was discussed within the OMEC-2 study to investigate the agreement in the definition and treatment of oligometastatic esophagogastric cancer in European-experienced centers. There was consensus that the definition of oligometastatic applies to cancers with only 1–2 metastatic lesions in either the liver, lung, retroperitoneal lymph nodes, adrenal gland, soft tissue or bone. Furthermore, there was consensus that oligometastatic disease can only be considered when the disease is at least stable after a median of 18 weeks of systemic therapy. However, there was no unity regarding the optimal treatment in the case of oligometastatic disease. Additionally, the cases discussed varied in terms of location and number of metastases, histology and time of detection, synchronous vs. metachronous. The results of OMEC-2 show even more that there is a big need for data of a randomized controlled trial in a clearly defined population of patients with only synchronous limited metastatic burden in a pure adenocarcinoma histology group without a mix of histology [20].

A systematic review and pooled analysis by Markar et al. [45], including trials published between 1990 and 2015 with a minimum of 10 patients with liver surgery for metastatic gastric cancer with hepatic-only disease and OS as primary outcome, investigated the influence of surgical resection of hepatic metastases regarding long-term survival, morbidity and mortality for gastric cancer patients. Other factors, e.g., multiple vs. single hepatic lesions or metachronous vs. synchronous disease, were also investigated regarding their influence on survival. The analysis included 39 studies with a median of 21 (range 10 to 64) liver resections. Liver surgery was associated with a median 30-day morbidity of 24% (0% to 47%) and mortality of 0% (0% to 30%). The median survival after 1, 3 and 5 years was 68%, 31%, and 27%, respectively, and even better in the Far East compared with trials in the Western world at 1 year (73% vs. 59%), 3 years (34% vs. 24.5%) and 5 years (27.3% vs. 16.5%). Overall, hepatic resections improved overall survival significantly (HR = 0.50; *p* < 0.001), especially in cases of single liver lesions compared with multiple hepatic metastases (odds ratio = 0.31; *p* = 0.011) [45]. This once again demonstrated that after appropriate selection of patients, surgery could be useful in a well-defined subgroup of patients with low tumor burden. In particular, patients with liver-limited disease could be another special subgroup.

There is a Chinese consensus on the diagnosis and treatment of gastric cancer with liver metastases, dividing into Type I resectable, Type II potentially resectable and Type III unresectable.

Type I shows 1–3 liver metastases; maximal diameter ≤ 4 cm or limited to one liver lobe without involving important vessels or bile ducts with technological resectability judged by a hepatobiliary surgeon. Type II liver metastases according to the classification are out of the range of Type I, with potential technological resectability and Type III is further divided into Type IIIa and b. Type IIIa shows multiple diffusely distributed metastatic lesions in both lobes without extrahepatic metastases and IIIb shows extrahepatic metastases in one or more organs with or without peritoneal carcinomatosis. Regarding the primary, Type I tumors are T ≤ T4a, lymph node metastases are within the D2 lymph node dissection area (not including Bulky N2) whereas Type II primaries include T4b or Bulky N2 or Bulky No. 16a2, b1. Type III include primary lesion directly and considerably invade adjacent tissues or organs; regional lymph nodes such as mesenteric lymph nodes or paraaortic lymph nodes fixed, fused, or not resectable, as confirmed by imaging examinations or biopsy.

The Chinese consensus recommends preoperative systemic treatments in Type I patients. In Type II patients, surgical resection is only recommended when R0 resection is intended. In Type III, cytoreductive surgeries are not encouraged [46].

The observational registry study of the Spanish AGAMENON national registry [47] showed data for surgery of metastases for esophagogastric cancer in real-world situations.

AGAMENON analyzed data from 1792 subjects in 32 centers, of whom 5% received surgical resection of metastases for esophagogastric cancer. The most frequent sites of metastatic resections were carcinomatosis of the peritoneum (29%), liver lesions (24%) and metastases of extra-regional lymph nodes (11%). Patients who were selected for resection of metastases showed improved survival rates, HR 0.34 (95% CI, 0.06–0.80, *p* = 0.021) [47]. Nevertheless, the median OS after surgery was 16.7 months (95% CI, 12.5–22.4) which is otherwise in line with modern systemic therapies shown by, e.g., the TOGA or Checkmate 649 trials [48,49], Table 2.

The 1- and 3-year relapse rates following R0 resection were high, with 58% and 65%, respectively, but interestingly the median time to recurrence since complete resection of metastases was up to 8.4 months (95% CI, 7.6–23.7) and the duration of systemic therapy before surgery increased mortality (HR 1.04 [95% CI, 1.01–1.07] *p* = 0.009). The AGAMENON investigators concluded that subjects with limited metastatic disease, selected on a clinical basis, can benefit from early surgeries. However, data of highly selected patients should be interpreted with caution, because in terms on pure survival numbers they are numerically no better than data from pure systemic therapy studies without surgery, but have an additional risk of surgical morbidity and possible mortality.

## 5. Upcoming and Running Trials for Oligometastatic Esophagogastric Cancer

The currently recruiting SURGIGAST trial (NCT03042169) of the French FREGAT group aims to evaluate overall survival after palliative gastric resection plus chemotherapy versus chemotherapy only in stage IV gastric cancer with only one solid organ metastatic site or with more than one metastasis in only one organ. Patients randomized to surgery can receive additional subtotal or total gastrectomy, depending on the location of the primary tumor. According to the original protocol, standardized D2-lymphadenectomy and resection of metastasis was not required, reflecting the situation of the REGATTA trial. With regard to the German FLOT-5/RENAISSANCE trial, SURGIGAST amended their protocol to the RENAISSANCE trial inclusion criteria of definition of oligometastatic disease and allowed surgery of the primary and treatment of the metastatic site.

Currently, a Dutch group is also preparing the COSTA trial, evaluating the potential benefits of surgery in limited metastatic disease as with SURGIGAST or RENAISSANCE, but COSTA has several limitations. COSTA is a multicenter, single-arm phase II study in pts undergoing palliative chemotherapy plus palliative resection to evaluate OS/PFS/HRQL synchronous metastatic gastric cancer treated with resection of only primary in addition to chemotherapy.

The COSTA idea is to evaluate the role of palliative resection, an issue already addressed by the REGATTA trial [27], so the oncologic role of pure palliative resections is quite well defined. The safety of non-curative gastrectomy for advanced gastric cancer is also well addressed by a big Dutch retrospective analysis of the DUCA group [58], evaluating all patients who underwent both non-curative gastrectomy (*n* = 115) and curative resection (*n* = 2087) in the Netherlands between 2011 and 2016. Although postoperative mortality was higher after non-curative surgery (9.6 versus 4.8%, *p* = 0.026), after propensity score matching there was no difference between groups (9.6 versus 7.0%, *p* = 0.415). Palliative surgery does not result in an additional risk of postoperative morbidity compared to curative gastrectomy. The oncological role as well as the safety aspects of a pure palliative gastric resection are therefore well defined and need no further evaluation [58].

The factors and variables discussed above should be addressed in modern OMD trials. In particular, the potential benefit from complete cytoreductive curatively intended surgery in cases with tumor regression after initial cycles of systemic induction chemotherapy needs further investigation.

## 6. Systemic Therapy in Oligometastatic Esophagogastric Cancer

A palliative systemic therapy in a metastatic or even oligometastatic gastroesophageal disease is standard and substantiated by the good data of the combination of chemotherapy with new antibodies. Furthermore, the role of perioperative systemic therapy is also a standard of care in the current entity. There is therefore a strong rationale for discussing whether (pseudo-) curation is the goal to combine the standardized most effective systemic therapy components with radical local procedures.

The ideal regimen should be based on 5-FU and platinum substances. Fluoropyrimidine plus platinum-based systemic therapy is a standard first-line regimen for unresectable advanced or metastatic human epidermal growth factor receptor 2 (HER2)- negative gastric and GEJ adenocarcinoma and should be part of the standard backbone in the limited metastatic disease [59,60,61,62,63,64,65].

The cytotoxic drug docetaxel has shown efficacy in metastatic settings, in first-line (docetaxel, cisplatin and fluorouracil-DCF) and second-line situation (docetaxel monotherapy) therapy [66].

Several phase II studies demonstrated that the addition of docetaxel to the combination of fluorouracil, leucovorin and oxaliplatin (FLOT) was more tolerable than the parent DCF regimen for metastatic and locally advanced gastric and GEJ adenocarcinoma with FLOT. In addition, the FLOT regimen elicits a stronger tumor response than other combinations, including the anthracycline-based triplets in locally advanced, resectable tumors, which were substantial in a preoperative situation [25,67,68]. In locally advanced, resectable gastric or GEJ adenocarcinoma, perioperative FLOT was able to improve overall survival compared with perioperative ECF/ECX and nearly doubled survival rates compared with pure surgery [26]. FLOT is one standard option in the oligometastatic disease.

The TOGA trial established the addition of the HER-directed antibody trastuzumab to chemotherapy with fluoropyrimidine plus cisplatin-based systemic therapy in patients with HER2-positive status in the 1st line palliative treatment of esophagogastric adenocarcinoma [48].

The non-interventional HERMES trial [69] followed the course of HER2-positive metastatic esophagogastric adenocarcinomas, under therapy with HER2-addressed trastuzumab-containing regimens, according to physicians’ choice. HERMES was able to show that, under a daily routine situation, not only in-label doublet cisplatin and 5-fluorouracil could be used as combination partners for trastuzumab. There was high diversity of backbones used; in particular, oxaliplatin-based doublets or FLOT as triplet were used in one third of patients.

The HER-FLOT regimen was also found to be safe, with a promising pCR rate of >20% achieving pathological complete response (pCR) [70,71]. Furthermore, the German phase II study PETRARCA showed recently that the addition of trastuzumab and pertuzumab to perioperative FLOT is feasible and significantly improved pCR compared to FLOT only [72].

In the clear palliative 1st line treatment situation of esophagogastric adenocarcinoma, nivolumab was the first programmed cell death (PD-1) inhibitor to demonstrate superior OS, along with a clinically meaningful progression-free survival (PFS) benefit and improved and durable objective responses in combination with fluoropyrimidine plus oxaliplatin-based chemotherapy compared to chemotherapy alone in previously untreated patients in a phase III situation, thus establishing a new standard first-line treatment mainly in PD-L1-positive patients according to their combined positivity score (CPS) [49]. In addition, the KEYNOTE-590 study established a further PD-1-addressed new standard of care with treatment of pembrolizumab in addition to chemotherapy in patients with previously untreated, advanced esophageal squamous cells as well as adenocarcinoma regardless of histology, with PD-L1 CPS of 10 or more [73].

Thus, the shown high response rates of fluoropyrimidine plus platinum-based chemotherapy plus docetaxel and/or PD-1 or HER-2-directed therapy [25,26,49,70,72] make the combinations attractive especially in the limited metastatic disease to enable a possible surgical R0 resection.

An interim analysis of the KEYNOTE-811, a randomized, global phase III study evaluating trastuzumab plus platinum and fluoropyrimidine doublet chemotherapy +/- pembrolizumab for unresectable or metastatic HER2-positive esophagogastric adenocarcinomas revealed that addition of the checkpoint inhibitor significantly induces tumor shrinkage and improves objective response rate [52]. This concept may also be an extremely attractive option to evaluate its use in the limited metastatic disease in combination with local therapies (see Table 2).

In contrast to other trials in the limited metastatic situation, the German RENAISSANCE/FLOT-5 trial [23] is based on three major aspects that, based on the data obtained in the FLOT-3 trial, are considered fundamental to the evaluation of the role of curative intended surgery in limited metastatic gastric cancer: (1) a good selection of gastric cancer patients who may still benefit from a curative intended treatment, even if they are already formally in a stage IV disease; (2) the selection includes patients’ factors, e.g., ECOG status and tumor-related factors including localization and extent of the oligometastatic situation; (3) the need for an effective chemotherapeutic regimen in the “neoadjuvant” situation to maximize response and in the “adjuvant” situation to stabilize surgical results. In addition, the molecular basis of the tumors, e.g., HER2 or PD L1 status, is also considered. It is important to clearly define that the aim of surgical procedure in modern trials is curatively intended in the oligometastatic setting, as FLOT-5 currently is [23]. Furthermore, induction with the most effective regimen, respecting the aggressive nature of esophagogastric adenocarcinomas, is important. Administration of only a delayed-effective systemic therapy after a local procedure would deny the fact that the limited metastatic situation is already a stage IV disease.

Even if the FLOT-5 trial does not show a benefit from curative intended surgery, this trial will still set a landmark by answering the question of pure systemic palliative intended vs. curative treatment by combining systemic and local therapies in a limited or oligometastatic gastric cancer disease.

For the future, we will need a strong selection of patients as, in the definition of oligometastatic disease in the FLOT-3 and FLOT-5 trials and in this well-defined subgroup, randomized comparison between surgery and palliative 1st line systemic therapy is needed.

## 7. Conclusions

Only a small number of esophagogastric cancers patients in formal stage IV of the disease show an oligometastatic or limited disease still with no uniform definition of what oligometastatic means in gastric cancer. Curatively intended treatment approaches in OMD esophagogastric cancers are still an experimental approach and the question remains unanswered as to whether these patients are still candidates for curative concepts because, until now, we have not had sufficient data from randomized phase III trials. The increasingly effective systemic therapies, such as triplet FLOT therapy or the addition of targeted therapies such as HER2-targeted therapy, anti-angiogenic therapy and/or the addition of checkpoint inhibitors, confront us with new questions. Do we still need local therapies such as surgery or radiotherapy when we have an increasingly effective system therapy, or should these methods be integrated all the more in order to combine all possible effective weapons for a maximum of tumor reduction?

However, one question must be asked at the start of therapy in the treatment of OMD patients in real life situations as well in the design of studies for OMD esophagogastric cancer patients. If the treatment is intended to be formally curative and life-prolonging, then the patient must be given a maximum of therapy with the aim of maximum tumor-/cytoreduction, or if the orientation of the treatment is clearly palliative, then compromises can be made, since the aim of local therapies in this context is only sparing of systemic therapy and symptom control.

## Figures and Tables

**Figure 1 cancers-14-05200-f001:**
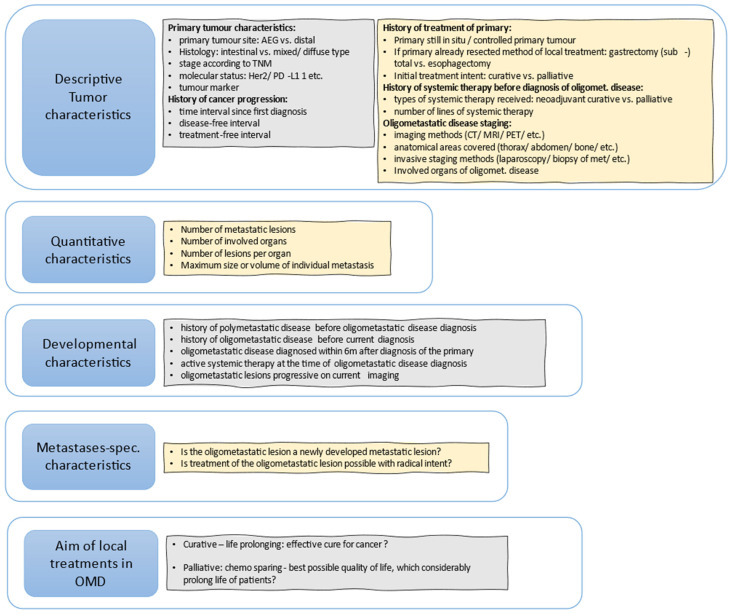
Useful questions to further define OMD, based, with modifications, on (Guckenberger et al., 2020) [18].

**Figure 2 cancers-14-05200-f002:**
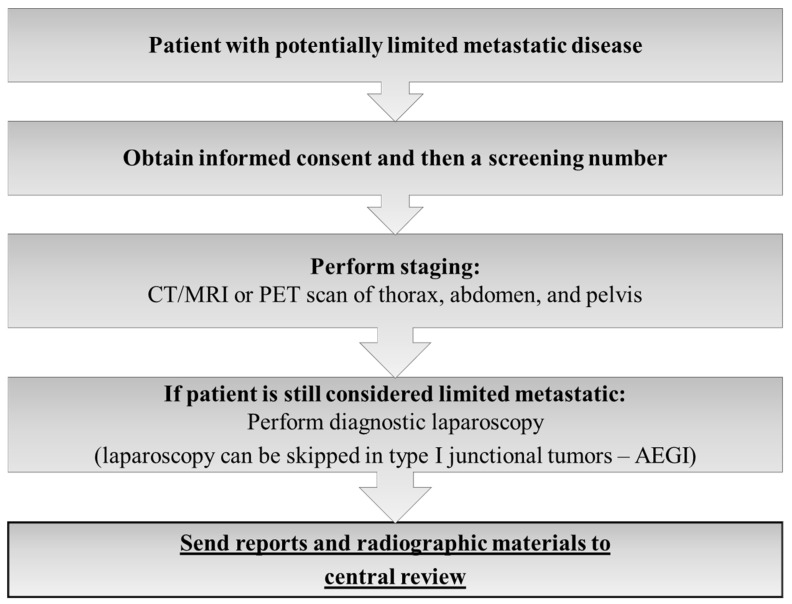
Information for decision-making used by the FLOT-5 central review expert panel.

**Table 1 cancers-14-05200-t001:** Definition of limited/oligometastatic disease in gastroesophageal cancers according to FLOT-5 vs. FLOT-3 trial.

FLOT-5 Definition of Limited/Oligometastatic	FLOT-3 Definition of Limited/Oligometastatic
Metastatic infestation of retroperitoneal lymph nodes (e.g., para-aortal, intra-aorto-caval, parapancreatic or mesenterial lymph nodes) only (Note: duodenum-invading gastric cancer, infestation of retropancreatic lymphatic nodes is not regarded as M1- disease) or/and infestation of a maximum of one organ combined with or without positive retroperitoneal lymph nodes according to the following criteria:	abdominal, retroperitoneal lymph node metastases only (e.g., para-aortic, intra-aortic-caval, peripancreatic, or mesenterial lymph nodes) or one incurable organ site with or without retroperitoneal lymph node metastases;
I. Localized potentially resectable peritoneal seeding according to stage P1 of the “Japanese Research Society for Gastric Cancer“- classification; also allowed within the trial, a so-called Peritoneal Cancer Index (PCI) according to Sugarbaker definition of ≤ 6 (of note, patients with a Japanese P-score 2 or 3, but PCI ≤ 6 are still eligible for trial inclusion) or	localized peritoneal carcinomatosis (P1 or P2 score), according to the classification of the Japanese Research Society for Gastric Cancer was allowed and considered 1 incurable organ site
II. Liver: not more than 5 metastatic lesions with the potency to be surgical R0-resectable or	fewer than five liver metastases, if the single organ site is the liver;
III. Lung: potentially surgically curable unilateral involvement or	
IV. Uni- or bilateral ovarian Krukenberg metastatic lesions in the absence of a further macroscopic peritoneal seeding or	Bilateral or unilateral Krukenberg tumors were allowed and considered 1 incurable organ site
V. Uni- or bilateral metastases of the adrenal gland or	Unilateral or bilateral adrenal gland metastases were also considered 1 incurable organ site
VI. Extra-abdominal metastatic lymph node seeding, e.g., supraclavicular, or cervical lymph node infestation or	Extra-abdominal lymph node metastases, such as supraclavicular lymph node involvement, were allowed and considered 1 incurable organ site.
VII. Localized bone infestation (respecting one radiation field) or	
VIII. Other metastatic infestation considered oligometastatic by the investigator and central review committee	

**Table 2 cancers-14-05200-t002:** Selected pivotal clinical trials in gastric and esophagogastric junction cancers [50].

Trial	Study Arms		Efficacy Outcomes		Clinical Implications	Ref.
		ORR	mPFS	mOS		
HER2						
ToGA *n* = 594 phase III trial, metastatic HER2+ G/GEJ cancers, first line	CTx (Capecitabine or 5-FU plus Cisplatin)	35%	5.5 months	11.1 months	Trastuzumab plus CTx is standard of care in first-line treatment in metastatic HER2+ disease.	[48]
	CTx (Capecitabine or 5-FU plus Cisplatin) with Trastuzumab	47%, *p* = 0.0017	6.7 months, *p* = 0.0002	13.8 months, *p* = 0.0046		
DESTINY-Gastric01 *n* = 188 phase II trial, HER2+ Asian metastatic gastric cancer patients, third or later-line	CTx (Irinotecan or Paclitaxel)	14%	3.5 months	8.4 months	FDA approval for Trastuzumab Deruxtecan in HER2+ G/GEJ cancer patients who have received a prior Trastuzumab-based regimen. No approval in Europe yet.	[51]
	Trastuzumab Deruxtecan	51%, *p* < 0.001	5.6 months, *p* = 0.01	12.5 months		
KEYNOTE-811 *n* = 264 phase III trial, metastatic HER2+ G/GEJ cancers, first-line, interim analysis	CTx (CAPOX or 5FU plus Cisplatin) plus Trastuzumab	51.9%	No data	No data	No mature data yet. Combination of HER2 targeting and immune checkpoint inhibition might have synergistic effects.	[52]
	CTx (CAPOX or 5FU plus Cisplatin) plus Trastuzumab with Pembrolizumab	74.4%, *p* = 0.00006	No data	No data		
VEGFR						
REGARD *n* = 355 phase III trial, metastatic G/GEJ cancers, second-line	placebo	3%	1.3 months	3.8 months	Ramucirumab mono therapy is approved for second-line treatment in G/GEJ cancers.	[53]
	Ramucirumab	3%, *p* = 0.76	2.1 months, *p* < 0.0001	5.2 months, *p* = 0.047		
RAINBOW *n* = 665 phase III trial, metastatic G/GEJ cancers, second-line	CTx (Paclitaxel)	16%	2.9 months	7.4 months	Ramucirumab in combination with Paclitaxel therapy is approved for second-line treatment in G/GEJ cancers.	[54]
	CTx (Paclitaxel) with Ramucirumab	28%, *p* = 0.0001	4.4 months, *p* = 0.0001	9.6 months, *p* = 0.017		
PD-1						
ATTRACTION-4 *n* = 724 phase II trial, metastatic Asian G/GEJ cancer patients, first-line	CTx (S-1 or Capecitabine plus Oxaliplatin)	47.8%	8.34 months	17.15 months	Biomarker-based patient selection is needed. In second-line treatment or later MSI-high G/GEJ cancers do benefit from immune checkpoint blockade.	[55]
	CTx (S-1 or Capecitabine plus Oxaliplatin) with Nivolumab	57.5%, *p* = 0.0088	10.45 months, *p* = 0.0007	17.45 months, *p* = 0.26		
ATTRACTION-2 *n* = 493 phase III trial, metastatic Asian G/GEJ cancer patients, third or later-line	placebo	0%	1.45 months	4.14 months		[56]
	Nivolumab	11.2%	1.61 months, *p* < 0.0001	5.26 months, *p* < 0.0001		
CheckMate-649 *n* = 1581 phase III trial, metastatic patients with oesophageal, gastric or GEJ cancers, first-line	CTx (Capecitabine plus Oxaliplatin or FOLFOX)	45%	6.1 months	CPS 5: 11.1 months; All patients: 11.6 months	FDA-approved Nivolumab in combination with chemotherapy as a first-line therapy in metastatic G/EGJ cancers. EMA decision is still pending.	[57]
	CTx (Capecitabine plus Oxaliplatin or FOLFOX) with Nivolumab	60%, *p* < 0.0001	7.7 months, *p* < 0.0001	CPS 5: 14.4 months, *p* < 0.0001; All patients: 13.8 months, *p* = 0.0002		
	Nivolumab plus Ipilimumab	No data	No data	No data		Not pub-yet

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
