# Peer review of "Perspectives on the Management of Oligometastatic Disease in Esophago-Gastric Cancer"

_cancers, 2022, doi:10.3390/cancers14215200_

Round 1
Reviewer 1 Report
The authors seek to summarize current evidence regarding treatment of metastatic and oligometastatic esophageal and gastric cancers. However the breadth of the summary is limited as there is minimal discussion of the role of consolidative radiotherapy to the primary site or regarding ablative radiation for oligometastatic disease despite adequate discussion of medical therapy and surgical resection of the primary tumor.
However, the manuscript would benefit from editorial assistance as there are extensive run on sentences and grammatical errors which make the article currently unsuitable for publication or extensive peer review. This is highlighted by inclusion of shorthand abbreviations (pts, line 178, line 412) which demonstrates a lack of proofreading prior to submission.
The review would benefit from a table outlining the enrollment criteria for FLOT-5 instead of as described in the text. Readers may also appreciate a figure (flow chart or table) summarizing current treatment recommendations for metastatic patients based on data as presented in CHECKMATE 649, KEYNOTE-062, and ATTRACTION-4 clinical trials. Furthermore, discussion of the definition of oligometastatic gastroesophageal cancer has been published from a European perspective (Kroese et al., 2022, EJC, PMID: 35134666) and should be included in the first section of the review.
Author Response
We thank the reviewer very much for the very valuable comments and the time invested to optimize the manuscript.
The role of radiotherapy in the oligometastatic disease of esophagogastric cancers is addressed and more intensively discussed. Literature from e.g. [Liu Qi, et al. Int J Radiat Oncol Biol Phys 2020;108:707–15 and Kroese TE, et al. Ann Surg Oncol (2022) 29:4848–4857 has been added.
A table [Table 1] with the definition of what is oligometastatic according to FLOT-5 has been added and in addition the old definition according to FLOT-3 trial was juxtaposed.
Also, a flow chart describing the decision tree used by the FLOT-5 central review expert panel for potential patient inclusion, has been added and illustrated by the new Figure 1.
Furthermore, a summarizing table [Table 2] of the current treatment landscape for metastatic patients based on data as presented in e.g., CHECKMATE 649, KEYNOTE-590, ATTRACTION-4, etc. clinical trials has been added.
The very informative paper on “Definitions and treatment of oligometastatic oesophagogastric cancer according to multidisciplinary tumour boards in Europe” reflecting a European perspective in the current problematic, published by Kroese T E et al., in European Journal of Cancer is now also discussed (Kroese et al., 2022, EJC, PMID: 35134666).
Manuscript has been checked by a native speaker.
Reviewer 2 Report
I would like to thank the Editor for giving me the opportunity to review this interesting manuscript. Thanks also to the authors for sharing their knowledge and effort in the field of esophagogastric cancer.
The present narrative review is very well structured and covers comprehensively different chapters that are currently debated in the treatment of limited stage IV esophagogastric cancer patients, even among experts in the field. Most of the recent evidence is cited and discussed in detail so that the reader is truly offered a good overview of the topic.
What first emerges from their study is that there is no clear definition of oligometastatic disease and most of the trials conducted to date used arbitrary criteria. Although I agree with the Authors about that, I think that for the specific subset of patients suffering from liver-only gastric cancer liver metastases some significant effort has been done in the attempt to define who is going to benefit the most from a combined approach based on perioperative systemic treatment and surgical resection. Indeed, in 2020 the Chinese Consensus on the diagnosis and treatment of gastric cancer with liver metastases (https://doi.org/10.1177/1758835920904803) proposed a classification system based on the likelihood of a surgical treatment being successful. Although the aforementioned consensus is based mainly on data coming from Eastern countries, I think it represents a good first step for a systematic approach to such an issue, and in my opinion, it should be discussed in this beautiful manuscript.
Moreover, just for the sake of completeness, I would suggest the Authors consider the inclusion of some references regarding several comprehensive meta-analyses recently published on the role of surgical resection and cytoreductive surgery + HIPEC in patients suffering from liver only (https://doi.org/10.1016/j.ctrv.2018.05.010; https://doi.org/10.1016/j.critrevonc.2021.103313) and peritoneal only metastases (https://doi.org/10.1245/s10434-022-12312-7; https://doi.org/10.1016/j.ejso.2021.05.016).
In conclusion, I think the present manuscript represents a meaningful, interesting read and after minor English language revisions, it will be suitable for publication in Cancers.
Author Response
We thank the reviewer very much for the very valuable comments and the time invested to optimize the manuscript.
A new figure [Figure 2] with useful questions for further description OMD in esophagogastric cancer has been included.
Reviewer 3 Report
You have done a wonderful job in this very interesting article but it would have been much nicer if there were charts/figures/schemes or tables explaining and describing the oligometastatic disease in Esophago-Gastric cancer since it is a new concept and we have to clarify it in the easiest way possible. The conclusions are confusing in terms of english comprehension in a way that the end results are hard to understand; run on sentences and incomplete sentences have to change.
Author Response
We thank the reviewer very much for the very valuable comments and the time invested to optimize the manuscript.
A brief discussion of the role of positive peritoneal cytology (CY+) is now implemented in the paper, including the role of e.g., peritoneal lavage respectively extensive peritoneal lavage (EIPL).
Also, the role of HIPEC in a multimodal approach is more intensively discussed and the PERISCOPE II- trial is addressed.
Manuscript has been checked by a native speaker.
Reviewer 4 Report
The authors should be congratulated on a comprehensive review of the management of oligometastatic disease in esophagogastric cancer: topic that represent un unmet clinical need for all GI-oncologists. The paper summaries a wealth of in-depth detail on the several papers aiming at conclusively define the concept of oligometastatic disease as well as up to date clinical trial results in this setting, highlighting their limitations due to the lack of statistical power and the non-comparability in terms of study population due to the inclusion criteria heterogeneity. The potential curative options in the limited metastatic setting currently under investigation are well depicted and a notable emphasis was given to the Reinassance-FLOT 5 study.
I have only some minor comments:
-We appreciate that the authors are not writing in their first language but there are unusual sentence structure throughout the text as well as punctuation errors. Whilst they do not overly distract from the message of the paper review of the syntax would be beneficial, especially in the following paragraphs: lines 217-224; 275-291; 370-380; 389-392; 393-408.
-The section on the role of peritoneal metastases in esophagogastric cancer would benefit from a brief discussion of the role of positive peritoneal cytology (CY+), in the absence of visible peritoneal implants and other metastatic sites at baseline, on the therapeutic algorithm (…predictive and prognostic aspects, evidence for surgical treatment after preoperative chemotherapy administration with or without conversion to CY-, role of HIPEC in a multimodal approach, reasons for exclusion from Reinassance trial…). Of course, we are aware about the absence of strong evidence from the literature.
Moreover in the same section, regarding HIPEC, the ongoing randomised phase 3 PERISCOPE II trial might be cited
10.1186/s12885-019-5640-2
10.2196/resprot.7790
Author Response
We thank the reviewer very much for the very valuable comments and the time invested to optimize the manuscript.
The Chinese consensus on the diagnosis and treatment of gastric cancer with liver metastases according to the paper of Zhang K et al. in Therapeutic Advances in Medical Oncology 2020 is now added and discussed in the current context in the manuscript as well the role of HIPEC and Cy+ disease is more intensively discussed.
Manuscript has been checked by a native speaker.
Round 2
Reviewer 1 Report
The clarity of the writing has dramatically improved with editing assistance and the manuscript can be accepted after minor proofreading.
Reviewer 3 Report
Dear authors
I am satisfied with the recent draft.
Congrats